# Effect of 1,2,4,5-Benzenetetracarboxylic Acid on Unsaturated Poly(butylene adipate-*co*-butylene itaconate) Copolyesters: Synthesis, Non-Isothermal Crystallization Kinetics, Thermal and Mechanical Properties

**DOI:** 10.3390/polym12051160

**Published:** 2020-05-19

**Authors:** Chin-Wen Chen, Te-Sheng Hsu, Kuan-Wei Huang, Syang-Peng Rwei

**Affiliations:** Institute of Organic and Polymeric Materials, Research and Development Center of Smart Textile Technology, National Taipei University of Technology, No. 1, Sec. 3, Chung-Hsiao East Road, Taipei 10608, Taiwan; cwchen@ntut.edu.tw (C.-W.C.); hsutesheng1991@gmail.com (T.-S.H.); wewe85625@gmail.com (K.-W.H.)

**Keywords:** poly(butylene adipate), poly(butylene itaconate), 1,2,4,5-benzenetetracarboxylic acid, partially cross-linking, thermoplastic, copolyester

## Abstract

Unsaturated poly (butylene adipate-co-butylene itaconate) (PBABI) copolyesters were synthesized through melt polymerization composed of 1,4-butanediol (BDO), adipic acid (AA), itaconic acid (IA) and 1,2,4,5-benzenetetracarboxylic acid (BTCA) as a cross-linking modifier. The melting point, crystallization and glass transition temperature of the PBABI copolyesters were detected around 29.8–49 °C, 7.2–29 °C and −51.1 and −58.1 °C, respectively. Young’s modulus can be modified via partial cross-linking by BTCA in the presence of IA, ranging between 32.19–168.45 MPa. Non-isothermal crystallization kinetics were carried out to explore the crystallization behavior, revealing the highest crystallization rate was placed in the BA/BI = 90/10 at a given molecular weight. Furthermore, the thermal, mechanical properties, and crystallization rate of PBABI copolyesters can be tuned through the adjustment of BTCA and IA concentrations.

## 1. Introduction

Aliphatic polyester, consisting of different lengths of CH_2_ group and ester bonds within the main-chain or side-chain, has been attracted more and more attention in recent years due to its lower melting temperature, reusable, flexibility, thermoplastic, easy processability and biodegradability [1,2,3,4,5,6]. Nevertheless, the thermal and mechanical characteristics of aliphatic polyesters can be enhanced via copolymerizing with different monomers to their suitability for a variety of commercial applications [7,8]. Unsaturated aliphatic polyester has been amended to practice the partial cross-linking via UV curing processing to adjust their physical property or through the cross-linking process with covalent bonds to produce thermoset-like copolyesters [9,10,11,12], or by combining it with epoxy materials [13,14,15]. The purpose of this operation is to achieve tremendous physical-chemical properties that provide unique characteristics of copolyesters, such as stronger hardness, higher rigidity, better tensile strength, chemical resistance and thermal stability [16,17,18].

Recently, copolyesters are developed rapidly due to the synthesis of new bio-based monomers and polymers. 2,5-furandicarboxylic acid is a popular bio-based aromatic diacid monomer consisting of a furan ring to replace the terephthalate acid, achieving an excellent gas barrier property [19,20,21,22,23]. Otherwise, the cross-linking modifiers are frequently undertaken to improve the thermal and mechanical properties of bio-based copolyesters [24,25,26,27]. By doing so, the bio-based aliphatic polyester can be formed a partial cross-linking network architecture by a small amount of multi-arm cross-linking modifiers, such as benzene-1,3,5-tricarboxylic acid and glycerol with the tri-arms; 2,2-bis (hydroxymethyl) 1,3-propanediol, ethylenediaminetetraacetic acid and 1,2,4,5-benzenetetracarboxylic acid with the tetra-arms; cyclohexane-1,2,3,4,5,6-hexacarboxylic acid and hexahydroxycyclohexane with the hexa-arms, etc. All the multi-arms functional modifiers were carried out below 0.5 mole% to form partial cross-linking networking, for which the thermal and mechanical properties of copolyesters could be tuned via adjusting the modifier concentration, while still maintaining the processability of thermoplastics copolyesters. Chan et al. have been synthesized the poly (ethylene sebacate-*co*-ethylene adipate) (PESA) with benzene-1,3,5-tricarboxylic acid, indicating a high Young’s modulus from 140 to 200 MPa with varying adipate content due to partial cross-linking network created by the trimesic acid unit [24,25]. Hsu and coworkers developed a fully bio-based poly (butylene succinate-*co*-propylene succinate) (PBSPS) system with glycerol adjust the thermal-mechanical properties, revealing the PBSPS copolyesters could be transformed from elastic to rigid characteristics by increasing the glycerol concentration, and the elongation decreased from 800% to 20% [26]. Furthermore, Chen and colleagues developed a novel unsaturated aliphatic copolyester like poly (butylene adipate-*co*-butylene itaconate) (PBABI), which copolymerized with ethylenediaminetetraacetic acid (EDTA) to enhance the physical property, showing the cross-linking modifiers were the best choice to form the steric network structure to improve the mechanical and thermal properties of these kinds of aliphatic copolyesters [27,28]. Otherwise, another interesting tetra-functional chemical is 1,2,4,5-benzenetetracarboxylic acid (BTCA), so-called pyromellitic acid, which plays the same role as a cross-linking modifier to control the crystallization rate and increase the enthalpy of the melting point due to the benzene ring in the center of BTCA [29,30]. Furthermore, Karki et al. proposed the potential use of the BTCA as an attractive multivalent template to act as a switch in H-bonds supramolecular networking, which could be performed to create novel photoactive arrangement applications [31]. As mentioned above, the aliphatic polyester can form a partial cross-linking architecture by tri- or tetra-arms cross-linking modifiers to investigate the thermal, mechanical properties and crystallization kinetics behavior.

Herein, the straightforward preparation of a series of PBABI copolyesters with BTCA via melting polymerization was synthesized in this research. The BTCA modifier within copolyesters could be improved the thermal stability and maintained mechanical properties via a partial crosslinking structure. Different compositions of AA/IA were conducted to tune a melting point range around 30–50 °C and at a constant BTCA ratio of 0.1 mole% to generate the mechanical strength. Furthermore, the non-isothermal crystallization kinetics was carried out to examine the crystallization rate for PBABI copolyesters, exhibiting the fastest crystallization rate was placed in a BA/BI ratio of 90/10. Finally, the influence of unique ratios of BTCA between 0.05 to 0.2 mole% at a fixed ratio of BA/BI = 90/10 was also explored to observe fluctuations in thermal and mechanical properties and to accommodate the appropriate BTCA concentration in the PBABI copolyesters.

## 2. Experimental

### 2.1. Materials

The adipic acid (AA, 99.8%) was supplied from Asahi Kasei Corporation. Itaconic acid (IA, 99%) and 1,4-butanediol (BDO, 99%) were purchased from the First Chemical Corporation (Taipei, Taiwan). The 1,2,4,5-benzenetetracarboxylic acid (BTCA, 97%) was obtained from Vetec. The 4-Methoxyphenol (99%), titanium butoxide (Ti(OBu)_4_, 97%), deuterium trifluoroacetic acid (d-TFA, 99.5%) and hexafluoroisopropanol (HFIP, 99%) were provided from Aldrich (t. Louis, MO, USA). The dibutyltin dilaurate (DBTDL, 95%) was attained from Alfa Aesar (Haverhill, MA, USA). All the chemicals were implemented in bulk polymerization via a 2 L reactor without any purification.

### 2.2. Sample Preparation

PBABI copolyesters with BTCA were copolymerized via one-pot melt polymerization. First, AA, IA and BTCA were esterified with BDO, 4-Methoxyphenol as an inhibitor, DBTDL and Ti(OBu)_4_ acting as a catalyst. A 2 L reactor was adopted in bulk copolymerization. The PBABI copolyesters represented to as BA/BI = x/y correspond to the ratio of butylene adipate/butylene itaconate, where x and y are the molar ratio of AA and IA. The molar ratio of [OH]/[COOH] was maintained at a molar ratio of 1.2/1 in all synthesized PBABI copolyesters, and the different molar ratios of BA/BI = 100/0, 95/5, 90/10, 85/15, 80/20 were chosen. The 4-Methoxyphenol, DBTEL and Ti(OBu)_4_ were set in 0.5 wt%, 1 wt% and 0.5 wt%, respectively. The concentration of the BTCA with different BA/BI ratios was carried out at 0.1 mole%. To consider the influence of various BTCA concentrations, the BA/BI ratio was fixed at 90/10 with the BTCA concentrations of 0.05, 0.1 and 0.2 mole%. The exact synthesizing conditions of the 2 L reactor have been described clearly in our previous report [27], and the detailed synthesis route was represented in Scheme 1.

### 2.3. Measurements

#### 2.3.1. Nuclear Magnetic Resonance Spectroscopy (^1^H NMR)

^1^H NMR spectrometer (JEOL ECZ600R 600 MHz, Tokyo Japan) was employed to identify the chemical structure of the synthesized PBABI copolyesters. At first, 100 mg of the synthesized copolyester was dissolved in d-TFA of 1 mL; then, the solutions were then transferred into 5 mm ^1^H NMR sample tubes. All the analyses were performed at room temperature and achieved 128 scans to collect the data for further analysis.

#### 2.3.2. Fourier Transform Infrared Spectroscopy (FT-IR)

FTIR Spectrometer (PerkinElmer, Waltham, MA, USA) was performed to measure the synthesized PBABI copolyesters in attenuated total reflection mode with an average signal of 32 co-added scans at a resolution of 4 cm^−1^ over a wavenumber range of 400–4000 cm^−1^.

#### 2.3.3. Intrinsic Viscosity (I.V.)

The synthesized PBABI copolyesters were measured in 1.0 g dL^−1^ within a mixture of phenol and tetrachloroethane in a weight ratio of 60/40 was obtained using an Ubbelodhe viscometer at 25 ± 0.05 °C. The IV of each sample was calculated using the Solomon-Ciuta equation: (1)$$\left[\eta \right]=\frac{\sqrt{\left\{2\left[\frac{t}{t0}-\ln\left(\frac{t}{t0}\right)-1\right]\right\}}}{C}$$
where C is the concentration of the solution; *t* is the flow time of the sample solution, and *t*_0_ is the flow time of the pure solvent. Measurements of each sample were made five times to calculate the averaged IV values and tabulated in Table 1.

#### 2.3.4. Gel Permeation Chromatography (GPC)

The Viscotek GPC System (1122 pump, 2707 Auto-Injector, 270 LS Laser Light Scattering Detector/Viscometer, Shodex 71 RI detector, OmniSEC 4.6 Station, Malvern, UK) was used with the HFIP 806 M Shodex column (Size: diameter and length in 8.0 mm and 300 mm, particle size is 10 μm). The synthesized PBABI copolyesters were dissolved in HFIP for 12 h and then filtered through a 0.2 µm PTFE filter membrane. Then, the oven temperature, flow rate and analysis time were set at RT, 1 mL min^−1^ with HFIP and 60 min, respectively.

#### 2.3.5. Differential Scanning Calorimetry (DSC)

DSC instrument (Hitachi High Tech. DSC-7000, Tokyo, Japan) was employed to evaluate the *T*_m_, *T*_c_, Δ*H*_m_ and ΔH_c_ of the PBABI copolyesters. The measurement was set at a heating rate of 10 °C min^−1^ from −50 to 150 °C and held for 5 min to eliminate thermal history. Then, these samples were cooled to −50 °C at a cooling rate of 10 °C min^−1^ for the first cycle. After that, the second cycle of the reheating process from −50 to 150 °C at the same heating rate of 10 °C min^−1^. All the DSC measurements were applied in a nitrogen atmosphere with aluminum pans.

#### 2.3.6. Thermogravimetric Analysis (TGA)

TGA (Hitachi, STA 7200, Japan) was adopted to measure the degradation temperature of the synthesized PBABI copolyesters. The synthesized samples in the weight range of 5–10 mg were heated from 50 to 600 °C at a heating rate of 10 °C min^−1^ under a nitrogen atmosphere. The degradation temperature from the TGA curve was determined at 5% weight loss (T_d-5%_).

#### 2.3.7. Dynamic Mechanical Analyzer (DMA)

The DMA (Tech Max DMS 6100, Tokyo, Japan) was adopted to estimate the viscoelastic properties of the PBABI copolyesters, including storage modulus (E′), loss modulus (E″) and loss tangent (tan δ). The synthesized samples were tested in compression mode with 150 mN at a frequency of 1 H_z_ and a set temperature range from −150 to 0 °C at a heating rate of 10 °C min^−1^. The sample was prepared at a size of 30 mm in length, 10 mm in width and 2 mm in thickness.

#### 2.3.8. X-ray Diffraction (XRD)

The film was prepared by hot-melt mechanical compression with a temperature of 80 °C and a pressure of 50 Kgf cm^−2^ for 1 min; then the XRD pattern of the PBABI film was collected using a Malvern Panalytical X’Pert^3^ powder diffractometer (Malvern, UK) with Cu K_α_ radiation (λ = 0.154 nm) in 2θ from 10 to 40 degrees at room temperature with a scanning speed of 0.2° min^−1^.

#### 2.3.9. Tensile Test

The hot melt compression molding was performed to prepare dumb-bell shaped specimens of the synthesized PBABI copolyesters at 80 °C and 3 kg cm^−2^ in 2 min. Then, the measurements were taken at a crosshead speed of 50 mm min^−1^ using Cometech QC-508M2F (Taichung, Taiwan) based on the ASTM d638 Type IV (Length: 33 mm, width: 6 mm, thicken: 3 mm) standard to attain a stress–strain curve. The yield strength, elongation at break, and Young’s modulus were evaluated from the stress–strain curve. All the data were obtained on an averaged value of 5 specimens. The sample was prepared at a size of 30 mm in length, 10 mm in width and 2 mm in thickness.

#### 2.3.10. Non-isothermal Crystallization Kinetic Procedures

All the PBABI copolyesters were carried out with a DSC instrument (Hitachi High Tech. DSC-7000, Japan) calibrated with Indium. All measurements were performed under nitrogen gas and atmosphere, and the weight of all samples was chosen in 5 mg sealed within an aluminum pan. Subsequently, the samples were heated up from room temperature to 150 °C at a rate of 10 °C min^−1^, kept at this temperature for 5 min to eliminate any thermal history, then cooled down to −50 °C at a different cooling rate of 2, 5 and 10 °C min^−1^, respectively. The exothermal curves of heat flow as a function of temperature were recorded to proceed the non-isothermal crystallization kinetics evaluation.

## 3. Results and Discussion

### 3.1. The Effect of the BA/BI ratio of PBABA Copolyesters with a BTCA Concentration of 0.1 mole%

Figure 1 displays the chemical structure of synthesized PBABI copolyester at a ratio of BA/BI = 90/10 was confirmed by ^1^H NMR spectra, and the other ratios of BA/BI were presented in Appendix A. The resonance peaks of PBABI copolyesters were identified and assigned in H_1_ (1.877–2.124 ppm, 3, 4-CH_2_ of adipic acid), H_2_ (1.960–2.206 ppm, 2, 3-CH_2_ of 1.4-butanediol), H_3_ (2.643–2.889 ppm, 2, 5-CH_2_ of adipic acid), H_4_ (3.689–3.912 ppm, 2-CH_2_ of itaconic acid), H_5_ (4.404–4.648 ppm, 1, 4-CH_2_ of 1.4-butanediol), H_6_ (6.031–6.142 ppm, –C=CH_2_ of itaconic acid), H_7_ (6.295–6.737 ppm, –C=CH_2_ of itaconic acid) and H_8_ (6.895–8.320 ppm, –CH of the benzenic ring in BTCA. All the resonance peaks in ppm, the integral ratio, and the calculated composition of IA for PBABI copolyesters were listed in Appendix A. The ratios of C=C bond of the IA were calculated in 4.07%, 4.21%, 8.48% and 9.53% for BA/BI = 95/5, 90/10, 85/15 and 80/20, respectively, implying the C=C bond of the IA may possibly isomerize into monomethyl fumarate, which has a significantly lower reactivity and cannot be detected in the NMR spectrum [32]. Furthermore, the concentration of the C=C bond of IA can be preserved near 50% in each PBABI copolyester during melt polymerization at high-temperature of 230 °C [33,34], implying the 50% C=C bond of IA may be reacted to be a saturated C-C bond in forming a partial cross-linking structure, and the residual C=C bond could participate in the successive UV curing procedure to adjust the mechanical properties of the PBABI copolyesters [27]. Brännström et al. studied the poly (butylene itaconate-*co*-butylene succinate) (PBIBSu) system via FT-IR analysis, which indicated the degree of curing of the PBIBSu copolyesters could be located around 50–75%, with various ratios of IA and SuA. Furthermore, the IA is not sufficient reactivity under melt polymerization, and the conversion of C=C in cross-linking formation could be made at 75% [35]. Consequently, staying near a 50% ratio for the C=C concentration of IA was critical for enhancing the unsaturated C=C to saturated C–C, to form a higher partial network architecture, which may decrease the backbone movement and the crystal region.

Figure 2 shows the FT-IR spectra of synthesized PBABI copolyesters, for which the absorption peaks related to the asymmetry and symmetry C–H stretch have been identified at 2954 and 2876 cm^−1^, respectively. The stretching vibration of the C=O in the ester bond at 1724 cm^−1^, a C–H bend absorption peak at a value of 1460 cm^−1^, and a band of 1254 cm^−1^ were correlated to C–O of the ester bond. The out-of-plane of the benzene ring within BTCA has a feature peak of around 745 cm^−1^. The most significant peak was discovered and remarked in 1639 and 817 cm^−1^, which was corresponding to the C=C bond stretching vibration within IA. Hence, the absorption peaks in 1639 and 817 cm^−1^ can be enhanced as raising the of IA concentration, explaining that the IA molecule was successfully copolymerized into a main-chain of PBABI copolyesters. Additionally, the C=C bond of IA was protected by radical inhibitor, 4-methoxyphenol, even though melt polymerization at a high temperature of 230 °C, with further UV curing, to control the mechanical properties of the PBABI copolyesters [27,32]. Tang et al. found that IA-based copolyesters could form a partial cross-linked structure through C=C of IA, using methyl methacrylate as an initiator to improve the mechanical property, after irradiation via the UV curing process [36].

Molecular weights and I.V. values of PBABI copolyesters were tabulated in Table 1. The I.V. values of PBABI copolyesters were valued at 0.75, 1.17, 1.23, 1.27 and 1.25 dL g^−1^ with gradually increasing the IA content by 5 mole%, suggesting chain entanglement can be enhanced by raising the IA concentrations to elevate the I.V. value. Moreover, the M_n_ shows a wide range of 16,787 to 39,024 g mole^−1^ implied that the polymerization degree of PBABI copolyesters could be improved with an increase of IA ratios to obtain a higher value of M_n_ under identical synthesizing procedures. The M_w_ had considerable values ranged from 36,064 to 161,950 g mole^−1^, reflecting that the PBABI copolyesters may be formed more great globule structures by a partial cross-linking structure in the presence of IA and BTCA. Furthermore, the polydispersion index values ranged from 2.15 to 4.15, resulting from a partial cross-linking structure for PBABI copolyesters. These results have a similar trend with the literature, and the PDI values are in a range of 1754 to 7.123 as an IA increased [24,25,26,27].

DSC cooling curves of the first cycle display in Figure 3a and reheating curve of the second-cycle in Figure 3b indicated the crystallization in PBABI copolyesters occurred during cooling procedures, and the melting behavior observed in the subsequent reheating at a rate of 10 °C min^−1^. A single crystallization peak was noticed clearly around 7.2–29 °C at different BA/BI ratios during the cooling process, suggesting that the molecular chain of PBABI copolyester was easy to pack into the ordered phase [36]. A broader crystallization peak was achieved when the IA concentration was increased to 20 mole%, implying a higher IA content could hinder the molecular chain from packing into the ordered state, and disrupt the crystallization region of the AA-rich domain. All the thermal properties are shown in Table 2, and the value of T_m_ reduced from 49 to 29.8 °C, as IA concentration increased to 20 mole%. Furthermore, After removing the thermal history, the melting peak of the PBABI copolyesters was shown to be in a continuous double peak, demonstrating that a competitive effect may exist in the crystallization zone of AA and IA, within the PBABI copolyesters, during melt polymerization [37,38]. The ∆*H*_m_ decreased from 57.3 to 45 mJ mg^−1^, as IA increased, indicating that the crystallization region can be disrupted in the presence of the IA molecule. Furthermore, all the thermal properties, degradation temperature, crystallinity and glass temperatures were decreased when the IA concentration increased.

Figure 4 presents the TGA profiles for PBABI copolyesters and the 5 wt% weight loss temperature (*T*_d-5%_) in Figure 4a occurred around 332.6–341.2 °C, indicating all samples have excellent thermal stability since the T_d-5%_ of all PBABI copolyesters was over 300 °C, and the *T*_d-5%_ values of PBABI copolyesters declined about 10 °C by increasing the IA content to 20 mole%, as can be seen in Table 2. Figure 4b displayed the derivative thermogravimetry (DTG), indicating a degradation speed was increased, and degradation temperature was decreased, as an increase of IA content. Hence, the thermal stability of the PBABI copolyesters could be diminished with an increase of IA concentration, as a result of the interruption of the crystallization area.

Figure 5 displays the DMA results of the PBABI copolyesters. The glass transition temperature (*T*_g_) was obviously obtained at the peak of tan δ, and observed a stable value range from −54.6 and −58.1 °C in the presence of IA, as shown in Figure 5a. The results suggested that the values of T_g_ were not correlated with the increase of IA concentration in the existence of BTCA. The BTCA as a modifier may play an essential role in maintaining the stability of the disordered region at relatively low temperature below −50 °C, even when the IA ratio was higher, due to the double-bond within IA, which can maintain the amorphous regime. The β-relaxation was also observed around −100 °C, which was correlated to the motion in the R-C=CH_2_ group, within IA, and the motion of the R-CH_2_ group within the molecular side-chain. As can be seen in Figure 5b, E′ decreased when the temperature was raised above *T*_g_ in the rubbery state because the double-bond within IA could be induced to disturb and damage the regularity of the molecular chain in the amorphous region, to decrease the E′ in the rubbery plateau region [39].

Figure 6 displays the XRD results of PBABI copolyesters in a 2θ range of 10–40°. The feature peak of XRD in all ratio of BA/BI was carried out around the 2θ values of 21.75°, 22.41° and 24.12° for the crystal lattices of (110), (020) and (020), respectively, which were related to the α-phase of virgin PBA [38,40,41,42,43,44]. Furthermore, all the XRD patterns of PBABI copolyesters in different BA/BI ratios have similar patterns compared to PBA, demonstrating that the crystal lattice of the BA unit could not be perturbed by the existence of the BI unit. Otherwise, PBABI copolyesters displayed crystallinity values in 42.4%, 41.6%, 40.3%, 39.9% and 34.2% for BA/BI = 100/0, 95/5, 90/10, 85/15 and 80/20, respectively, indicated sufficient IA content could interrupt chain packing into the ordered state to lowering crystallinity. Generally, the crystallinity of the PBABI copolyesters was around 40%, which specified that the crystal formation was favored by the overall van der Waals interaction and the BTCA, to form partial cross-linking polyesters. To the best of knowledge, PBABI copolyesters displayed an increase of chain flexibility with increasing BI content. This increased chain flexibility may have enhanced a more-effective rearrangement of the polymer chains, allowing van der Waals interactions to encourage a more significant degree towards the formation of the crystalline domain and may have increased the crystallinity. Unfortunately, the benzene ring of BTCA provided a crucial role in providing a robust steric hindrance to limit the chain rearrangement into the crystallization regime, revealing the crystallinity decreased with an increase of IA concentrations. Generally speaking, the calculated crystallinities obtained via DSC have similar trends than those from XRD results, suggesting a beneficial relationship between the two measurements.

The stress–strain curve was taken via tensile tests, including the yield strength, elongation at the break, and Young’s modulus of the PBABI copolyesters, as shown in Figure 7 and tabulated in Table 3. In the tensile test, the macroscopic deformation of BA/BI = 100/0 was established at the highest strength of 15.19 MPa and elongation at break of 56.13%, and all the mechanical properties decreased and exhibited a brittle characteristic when the IA concentration increased above 10 mole%. The yield strength, elongation and Young’s modulus at different BA/BI ratios were evaluated in a range of 5.81–15.19 MPa, 13.78–56.13% and 32.19–168.45 MPa, respectively, suggesting mechanical properties of PBABI copolymers was reduced with an increase of IA concentration due to decrease of the crystallinity. Similarly, Panic et al. have been proposed that an increase of the itaconate concentration could worsen mechanical properties and hinder the packing behavior of the molecular chain occurrence [45].

### 3.2. Non-Isothermal Crystallization Kinetics of PBABI Copolyesters

As can be seen in Figure 8, the non-isothermal crystallization curves of PBABI copolyesters with different ratios of BA/BI were recorded at different cooling rates of 2, 5, 10 °C min^−1^, exhibiting a single exothermic peak, which becomes broader and moves to lower temperatures as the cooling rate increased due to a different thermal equilibrium [46,47]. Increasing the heating rate from 2 to 10 °C min^−1^, the initial crystallization temperatures were declined, which related to the kinetic aspects of these conditions occurring in non-equilibrium materials.

#### 3.2.1. Non-Isothermal Crystallization Kinetics Based on Avrami Equation

The primary stage of non-isothermal crystallization could be well defined by the Avrami [48,49,50] in Equations (2) and (3)
(2)$$1-Xt\left(\%\right)=e^{-Kt^{n}}$$
or
(3)log[−ln(1−Xt(%))]=log(K)+n log(t)
where Xt (%) is the relative crystallinity, K is the crystallization rate constant, and n is the Avrami exponent, which is correlated to the growth geometry of crystals.
(4)$$Xt\left(\%\right)=\frac{\int_{0}^{t}\left(\frac{dH_{c}}{dt}\right)\;dt}{\int_{0}^{}\left(\frac{dH_{c}}{dt}\right)\;dt}$$
where the dH_c_ has represented the enthalpy of crystallization at the target temperature during the time interval dt via DSC measurement via Equation (4). The integral limits of t and ∞ are adopted to denote the time during the crystallization occurrence and the end of the crystallization procedure, respectively. Appendix A displayed the plot of relative crystallinity as a function of time at different cooling rates. All the experimental results were plotted with the log{−ln[1 − *X_t_* (%) ]} as a function of log(t) at the *X_t_* (%) ranged from 20–80% and were well fitted by Avrami equation at various [51,52,53]. (See Appendix A) All the Avrami parameters, n and K were linearly regressed to obtain from the slopes and intercepts of the curves for PBABI copolyesters at different cooling rates. The half-time (t_1/2_, min) and growth rate (G, min^−1^) were also calculated from Equations (5) and (6), and all the detailed information were tabulated in Table 4.
(5)$$t_{1/2}={\left(\ln\;\left(2\right)/K\right)}^{\left(1/n\right)}$$
(6)$$\;G=1/t_{1/2}$$

Figure 9 shows the growth rate as a function of IA content for PBABI copolyesters at different cooling rates. Obviously, the crystallization rate first increased and then decreased by raising the IA content within the PBABI copolyesters at each cooling rate. The highest value of growth rate was located at the IA content of 10 mole%, implying the crystallization rate could be enhanced with adding 10 mole% IA into PBABI copolyesters as a nucleating site, which would improve the flexibility of the molecular chain, thus reducing the energy barrier to drive the chain diffusion into the crystal lattice to promote crystallization behavior. Subsequently, the IA molecular could hinder the chain packing behavior and disrupt the regularity of the molecular chain in the IA concentration of more than 10 mole%. Further, the Avrami exponent of PBABI copolyesters is located around 3–5 and listed in Table 4, revealing that the crystal lattice tended to exhibit a spherical growth geometrical mechanism. The lowest half-time (t_1/2_) was placed in BA/BI = 90/10, implying the fastest crystallization rate was obtained at the same cooling rate, as a 10 mole% of IA inside the PBABI copolyesters.

#### 3.2.2. Non-Isothermal Crystallization Kinetics Based on Mo Equation

Mo and colleagues have been developed a technique by merging the Avrami [49] and Ozawa [54] model to explain exactly non-isothermal crystallization kinetics [55] and the Equation (7) as follows,
(7)$$\ln\phi =\frac{1}{m}\ln\left[\frac{K\left(T\right)}{Z}\right]-\frac{n}{m}\ln t$$

Here, the F (T) = [K (*T*)/Z] ^1/m^ and a = n/m, the Equation (7) can be modified to Equation (8):(8)lnϕ=lnF(T)−a lnt
where the physical meaning of F(*T*) is the cooling rate value chosen in each crystallization time at which the system has a defined degree of crystallinity, meaning a higher value of F(*T*) exhibits a lower crystallization rate.

The plot of log(ϕ) versus log(t) for PBABI copolyesters with a different BA/BI concentrations were presented in Appendix A and the value of F(*T*) and a can be obtained by intercept and slope via Equation (8), as listed in Table 5. The results of proposed Mo’s model that displayed an excellent linear relation could be well described as non-isothermal crystallization behavior for PBABI copolyesters.

Figure 10 presents the plot F(*T*) as a function of IA content with a crystallinity at a range of 0.2–0.8 for PBABI copolyesters. At first, the value of F(*T*) was reduced and then raised as the increase of IA content in PBABI copolyesters at a relative crystallinity, presenting the crystallization rate increased first and then decreased depended on the IA content. Moreover, the lowest value of F(*T*) is located in 10 mole% IA content, suggesting the BA/BI = 90/10 has the highest crystallization rate. These results of Mo’s model are well agreement with the Avrami model, as displayed in Figure 9.

Activation energy is a driving force to control the crystallization behavior of copolymers, which is associated with the energy required for the transport of crystalline chains across the inter-phase. The Kissinger model [56] is most implemented to evaluate the activation energy via the Equation (9) as follows,
(9)$$\frac{d\left[\ln\left(\emptyset /T_{p}^{2}\right)\right]}{d\left(1/T_{p}\right)}=-\frac{\Delta E}{R}$$
where T_p_ is the peak of crystallization temperature, ΔE is the activation energy, R is the gas constant. The slope of the Equation (9) was equal to -ΔE/R and plotted in Appendix A. After that, the value of ΔE as a function of IA content was plotted in Figure 11. It can be seen that the lowest ΔE value was detected in 10 mole% IA. The reduced activation energy is caused by the incorporating of a small amount of IA, which plays a role in determining the nucleating site within the PBABI copolyesters to accelerate the molecular chain packing. Nevertheless, as the IA concentration exceeded 10 mole%, the irregular molecular chain structure would be a dominant role, leading to rising the activation energy. This statement is also in good agreement with the Avrami’s and Mo’s results.

The nucleation activity (*ϕ*) of non-isothermal crystallization behavior could be calculated by the Dobreva model [57] in which B_s_, a parameter related to the nucleation behavior, can be estimated through following Equation (10),
(10)$$ln\phi =const.-\frac{B_{s}}{2.3\Delta T_{p}^{2}}$$
where Δ*T*_p_ = *T*_o_– *T*_p_, the *B*_s_ can be obtained from the slope of the plot of ln(ϕ) vs. 1/2.3Δ*T*_p_^2^ while ∅ can be calculated from *B*_s_, according to the following Equation (11),
(11)$$\phi =\frac{B_{s}^{*}}{B_{s}^{0}}$$
where *B*_s_^*^ and *B*_s_^0^ are the values of *B*_s_ for heterogeneous and homogeneous nucleations, respectively. As best of our knowledge, the nucleation activity decreased with inserting an additive. [57] The plot of ln *ϕ* vs. 1/2.3Δ*T*_p_^2^ for PBABI copolyesters is displayed in Appendix A, from which the results *ϕ* are calculated as listed in Table 6. For all BA/BI ratios, all the *ϕ* is below 1 and *ϕ* has a minimum value of 0.2896 at BA/BI = 90/10. These results suggested that the existed IA had facilitated the nucleation site formed for the PBABI copolyesters in an IA content of 10 mole% at a given molecular weight.

### 3.3. The Effect of Different BTCA Concentrations at BA/BI = 90/10 of PBABI Copolyesters

Appendix A exhibits the ^1^H NMR results of PBABI copolyesters with different contents of BTCA. The resonance peaks of PBABI copolyesters with different BTCA contents were identified and assigned in H_1_ (1.893–1.916 ppm, 3, 4-CH_2_ of adipic acid), H_2_ (1.971–1.998 ppm, 2, 3-CH_2_ of 1.4-butanediol), H_3_ (2.654–2.677 ppm, 2, 5-CH_2_ of adipic acid), H_4_ (3.678–3.706 ppm, 2-CH_2_ of itaconic acid), H_5_ (4.415–4.438 ppm, 1, 4-CH_2_ of 1.4-butanediol), H_6_ (6.085–6.102 ppm,–C=CH_2_ of itaconic acid), H_7_ (6.678–6.706 ppm,–C=CH_2_ of itaconic acid) and H_8_ (7.072–7.102 ppm,–CH of the benzenic ring in BTCA. All the chemical shift and integral values were recorded in Table 7, revealing the calculated BTCA concentrations were taken in 0.16, 0.17 and 0.22 mole% for a feed content of 0.05, 0.1 and 0.2 mole%, respectively.

Figure 12 shows the DSC traces for PBABI copolyesters to investigate the thermal response. As shown in Figure 12a, crystallization was initiated at a higher temperature with BTCA concentration of 0.05 mole%, and also had a larger ∆*H*_c_, implying the crystallization behavior was correlated with the amount of BTCA. The molecular chain could be driven to stack into a crystal regime due to the nucleating effect of a smaller amount of BTCA concentration. The subsequent reheating procedure was displayed in Figure 12b, and to measure the *T*_m_ and ∆*H*_m_. Then, the peak melting point was split into continuous double peaks, signifying the occurrence of competitive behavior concerning AA and IA in the crystallization region. To be more exact, an increase of BTCA concentrations can exhibit a substantial steric hindrance to limit the backbone of packing into a crystal regime owing to the coplanar conformation of BTCA, thus clearly decreasing the enthalpy of crystallization.

Figure 13 illustrates the DMA analysis of Tan δ and storage modulus in a fixed ratio of BA/BI = 90/10 at different concentrations of BTCA. It is manifest from Figure 13a that the Tan δ of PBABI copolyesters increased slightly at higher concentrations of BTCA, and the T_g_ values were located around −49.1–−54.9 °C, suggesting the steric hindrance of the BTCA could restrict the side-chain motion in the amorphous zone to improve the T_g_. As can be seen in Figure 13b, the largest storage modulus was observed at a BTCA ratio of 0.2 mole%, suggesting that the PBABI copolyesters in both glassy and rubbery region have a trend to be harder characteristic, attributed to the relatively higher fraction of BTCA. Alternatively, a higher BTCA concentration within PBABI copolyesters revealed the more coplanar benzene ring to produce a more robust partial cross-linking structure, which could improve the mechanical properties of the PBABI copolyesters.

Thermal property measurements are summarized in Table 8. At a BTCA content of 0.05 mole%, the onset point of T_c_ was reached at a temperature of 24.5 °C, indicating it could crystallize at a higher temperature and grew a better-sized spherulite than other concentrations of BTCA, which was also evident in the higher ∆*H*_c_ value of −56.5 mJ mg^−1^. The DSC trace of *T*_m_ in the reheating procedure had a similar trajectory in continuous double peaks around 40–50 °C, and the ∆*H*_m_ values were carried out 51.2, 47.9 and 38.2 mJ mg^−1^ for BTCA concentrations of 0.05, 0.1 and 0.2 mole%, respectively. It is interesting that the most considerable value of ∆*H*_m_ occurred at a BTCA ratio of 0.05 mole%, implying a small amount of BTCA concentration may improve the thermal behavior.

The absorption peak of FT-IR spectra in PBABI copolyesters at various ratios of BTCA is displayed in Appendix A. It is not surprising that the FT-IR curve of each ratio of BTCA in BA/BI = 90/10 exhibited a similar absorption position and mentioned in Figure 2. Otherwise, XRD patterns for BA/BI = 90/10 at BTCA concentrations from 0.05 to 0.2 mole% are presented in Appendix A. The values of 2θ of BA/BI = 90/10 with BTCA of 0.05–0.2 mole% of PBABI copolyesters were identified around 21.97°, 22.66° and 24.38°, which relates to the crystal lattices of (110), (020) and (020), respectively. All the intensity of crystal lattices decreased in a BTCA ratio from 0.05 to 0.2 mole%, which indicated that the degree of crystallization for PBABI copolyester was reduced by increasing the concentration of BTCA. The most considerable ∆*H*_m_ value was also observed at a ratio of 0.05 mole%. However, the intensity of feature peaks of (110) and (020) decreased gradually with the BTCA was increased to 0.1 and 0.2 mole%, implying the crystal region was reduced as the BTCA strengthened. More specifically, the higher concentration of BTCA can increase the steric hindrance to disrupt the crystal regime, and the crystal region becomes smaller in lowering the ∆H_m_. Hence, the BTCA acted an essential role in regulating the thermal behavior and mechanical property.

The stress–strain curves of PBABI copolyesters at different ratios of BTCA were measured, as illustrated in Figure 14 and summarized in Table 9. The yield strength was raised with an increase of BTCA concentration due to the more content of BTCA, and the elongation was decreased corresponding. The Young’s modulus could be enhanced dramatically when a small amount of BTCA was copolymerized into the PBABI copolyesters; hence, the mechanical property can be tuned through the adjustment of BTCA concentrations. However, the trend in stress deviation increased with an increase in the BTCA concentration, which could be ascribed to the higher partial degree of cross-linking. Fisher and colleagues have been studied unsaturated poly(propylene fumarate) after UV–Vis irradiation, proposed that the 3D network structure may well improve the tensile properties with different concentrations and types of cross-linking modifier [58].

## 4. Conclusions

A series of unsaturated PBABI copolyesters with the BTCA were systematically synthesized via bulk polymerization and identified the structure through ^1^H NMR and FT-IR analysis. The *T*_m_, T_c_ and *T*_g_ of the PBABI copolyesters were found in a range of 30–50 and 10–30, and −51–−55 °C, respectively, and has an excellent thermal property at a *T*_d-5%_ above 330 °C. The feature peak of XRD was located in the 2θ values of 21.75°, 22.41° and 24.12° for the crystal panels of (110), (020) and (020), respectively; then the crystallinity from the XRD pattern was decreased from 42.4% to 34.2% with gradually raising the BA/BI ratio to 20 mole% suggesting sufficient IA content could interrupt chain packing into the ordered state. The Young’s modulus can be tuned, ranging between 32.19–168.45 MPa at different BA/BI ratios. The highest crystallization rate was measured at the BA/BI = 90/10 by non-isothermal crystallization kinetics analysis. Furthermore, the BTCA played a crucial role in controlling the enthalpy of crystallization and a higher concentration of BTCA could increase the steric hindrance to disturb the crystal regime. The Young’s modulus could be dramatically enhanced by the increase of 0.1 mole% of BTCA at BA/BI = 90/10; hence, the PBABI copolyesters could be coated on the 3D air mesh fabric for a phase change material usage. Moreover, advantages of the PBABI copolyesters could offer biodegradable, low *T*_m_ and UV-Vis curable to utilize in smart textiles and tunable thermal behavior and mechanical property through the adjustment of IA content and BTCA concentrations.

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
