# Peer review of "Effect of 1,2,4,5-Benzenetetracarboxylic Acid on Unsaturated Poly(butylene adipate-co-butylene itaconate) Copolyesters: Synthesis, Non-Isothermal Crystallization Kinetics, Thermal and Mechanical Properties"

_polymers, 2020, doi:10.3390/polym12051160_

Round 1
Reviewer 1 Report
This manuscript presented an interesting work about the synthesis and characterization of Unsaturated Poly(butylene adipate-co-butylene itaconate) Copolyesters. The work has potential. However, some points, listed below, need to be improved.
Section 2.2: please better describe the conditions used during the synthesis, such as, time, temperature, pressure, agitation speed,… I addition, I suggest add a scheme to illustrate the modification process.
Page 6 Line 196: it is very hard to see the band at 1639 cm-1.
Page 7 Lines 215-216: I suggest add the PDI values obtained by others authors.
Table 2: please better discuss the all the results presented in Table 2.
Table 3 TGA results: I can not find a table with TGA results, as pointed by the authors in page 8 line 242. Add a table with TGA and DTG results and better discuss these results in the manuscript. Please check the number of all tables in the manuscript.
Page 10: the authors should correlate the mechanical properties presented in Table 3 with the crystallinity values obtained from DSC.
Page 11: better comment the results showed in Figure 8.
Page 12 Table 4: I suggest better discuss all the results presented in Table 4.
Author Response
Reviewer #1
Comments and Suggestions for Authors
- Section 2.2: please better describe the conditions used during the synthesis, such as, time, temperature, pressure, agitation speed,… I addition, I suggest add a scheme to illustrate the modification process.
Response: Thanks for the reviewer’s suggestion. We have added a scheme for synthesis route, chemicals, time, temperature, pressure, and agitation speed. Please see the revised manuscript in scheme 1.
- Page 6 Line 196: it is very hard to see the band at 1639 cm-1.
Response: Thanks for the reviewer’s recommendation. In order to clarify the figure, all the curves are modified in the thin line. Please see the revised manuscript.
- Page 7 Lines 215-216: I suggest add the PDI values obtained by others authors.
Response: The PDI values from the literature were modified in the revised manuscript.
- Table 2: please better discuss the all the results presented in Table 2.
Response: Thanks for the reviewer’s suggestion. We have deeply discussed the results of Table 2 and presented them in the revised manuscript.
- Table 3 TGA results: I can not find a table with TGA results, as pointed by the authors in page 8 line 242. Add a table with TGA and DTG results and better discuss these results in the manuscript. Please check the number of all tables in the manuscript.
Response: Thanks for the reviewer’s careful observation. We have a wrong typing, and the TGA results are displayed in Table 2. As the reviewer’s suggestion, we have put the DTG results in Figure 4(b) and discussed in the revised manuscript. Furthermore, we have checked the number of all tables in the revised manuscript, and thanks for the reviewer again.
- Page 10: the authors should correlate the mechanical properties presented in Table 3 with the crystallinity values obtained from DSC.
Response: Thanks for the reviewer’s comment. We have been correlated with the mechanical properties and the crystallinity values, and please see the revised manuscript.
- Page 11: better comment the results showed in Figure 8.
Response: Thanks for the reviewer’s suggestion. We have modified better comments in Figure 8, and please see the revised manuscript.
- Page 12 Table 4: I suggest better discuss all the results presented in Table 4.
Response: Thanks for the reviewer’s comments. We have modified the discussion in Table 4, and please see the revised manuscript.

Reviewer 2 Report
The authors reported systems based on unsaturated poly (butylene adipate-co-butylene itaconate) (PBABI) copolyesters. They studied detailed the thermal behavior, crystalization process and mechanical properties by varing the constituents' ratios (the sequences of adipic acid and itaconic acid) and the crosslinking agent, 1,2,4,5-benzenetetracarboxylic acid (BTCA).
The results are well organized and comparable with the literature. English needs to be very well revised as many of the phrases were not clear enough and various editorial mistakes were found consistently.
Additionally please consider some comments/ observations to improve the quality of the study:
- Phrase Page 1 Lines 30, 31 “Nevertheless, weaker thermal and mechanical characteristics have been limited in many ways 31 to their suitability for commercial applications ” was not clear enough. The authors are requested to reformulate to be better understood the information.
- Line 38 The authors inserted many times “bio-based” throughout the text and within the same phrase; They are requested to reformulate the phrases to sound better.
- Page 2 line 48 The authors claimed “All the multi-arms functional groups of cross-linking modifiers were examined in a small amount of molar ratio to form partial cross-linking” Some details regarding the minimum or maximum amount of cross-linking agents should be added.
- Page 2 Line 49 Authors claimed that “...thermal and mechanical properties could be precisely controlled...” Some additional information is needed to be provided.
- Phrase “The BTCA could be established a partial crosslinking structure and the coplanar steric conformation to ensure thermal stability and maintain mechanical properties.” was not clear. The authors might reformulate it so that the information do not be misspelled.
- The authors synthesized previously the Poly(butylene adipate-co-butylene itaconate) Copolyesters but what exactly is the contribution to the present paper to the study of these systems was not clearly mentioned; the authors are requested to state clearly the developments they made previously and the purpose of the newly developed system to avoid confusions.
- Section 2.3.3 The authors described some special conditions to measure the intrinsic viscosity but how these conditions were determined was not mentioned; additional details on how those conditions were established should be included in the description of the method.
- Sections 2.3.4 Details on what type and length of column was used were not included; the authors are requested to provide a full description of the method they used for GPC measurements.
- Page 6 line 195-196 Phrase “The intensity of these two absorption peaks can be enhanced as rasing the of IA concentration, explaining that the IA molecule was successfully copolymerized into a main-chain of PBABI copolyesters” was not clear and it should be reformulated.
- The authors did not conclude which one of the four compositions could be used successfully and practically as a smart coating material on 3-D air mesh fabric. This mention was done just within the abstract.
Author Response
Reviewer #2
Comments and Suggestions for Authors
- Phrase Page 1 Lines 30, 31 “Nevertheless, weaker thermal and mechanical characteristics have been limited in many ways 31 to their suitability for commercial applications ” was not clear enough. The authors are requested to reformulate to be better understood the information.
Response: Thanks for the reviewer’s comments. After carefully reading and understanding the literature, the sentence was modified to “Nevertheless, the thermal and mechanical characteristics of aliphatic polyesters can be enhanced via copolymerizing with different monomers to their suitability for a variety of commercial applications.” Please see the revised manuscript.
- Line 38 The authors inserted many times “bio-based” throughout the text and within the same phrase; They are requested to reformulate the phrases to sound better.
Response: Thanks for the reviewer’s suggestion. The sentence was rewritten in “Recently, copolyesters are developed rapidly due to the synthesis of new bio-based monomers and polymers. 2,5-furandicarboxylic acid is a popular bio-based aromatic diacid monomer consisting of a furan ring to replace the terephthalate acid, achieving an excellent gas barrier property”. Please see the revised manuscript.
- Page 2 line 48 The authors claimed “All the multi-arms functional groups of cross-linking modifiers were examined in a small amount of molar ratio to form partial cross-linking” Some details regarding the minimum or maximum amount of cross-linking agents should be added.
Response: Thanks for the reviewer’s recommendation. The sentence was modified in “All the multi-arms functional modifiers were carried out below 0.5 mole% to form partial cross-linking networking”. Please see the revised manuscript.
- Page 2 Line 49 Authors claimed that “...thermal and mechanical properties could be precisely controlled...” Some additional information is needed to be provided.
Response: Thanks for the reviewer’s comment. The sentence was rewritten in “the thermal and mechanical properties of copolyesters could be tuned via adjusting the modifier concentration, while still maintaining the processability of thermoplastics copolyesters.” Please see the revised manuscript.
- Phrase “The BTCA could be established a partial crosslinking structure and the coplanar steric conformation to ensure thermal stability and maintain mechanical properties.” was not clear. The authors might reformulate it so that the information do not be misspelled.
Response: Thanks for the reviewer’s suggestion. In order to clarify the meaning, the sentence was modified in “The BTCA modifier within copolyesters could be improved the thermal stability and maintained mechanical properties via a partial cross-linking structure.” Please see the manuscript.
- The authors synthesized previously the Poly(butylene adipate-co-butylene itaconate) Copolyesters but what exactly is the contribution to the present paper to the study of these systems was not clearly mentioned; the authors are requested to state clearly the developments they made previously and the purpose of the newly developed system to avoid confusions.
Response: Thanks for the reviewer’s comments and suggestions. At first, we synthesized the poly(butylene adipate-co-butylene itaconate) system in order to obtain low Tm copolyester and controllable hardness for coating on 3D mesh fabric. After that, we face new big trouble, that the solidation rate of copolyesters (from melt-state to solid) is quite slow (ACS Omega, 2020, 5, 3080-3089), implying the crystallization rate of previous low Tm copolyester needs to speed up for applications. After careful consideration of this situation, we think that the key issue is in the modifier. Hence, we try to improve and solve this problem; we have replaced the original modifier ethylenediaminetetraacetic acid (EDTA) to BTCA. By doing so, we obtained two advantages of fast crystallization rate and higher enthalpy of new modified copolyesters. This is why we have discussed the non-isothermal crystallization kinetics in this article and found the fastest crystallization rate in BA/BI = 90/10 with BTCA. After that, we adopted this new copolyester as phase change materials and blended it with poly(hexamethylene terephthalate) to produce the masterbatch. Finally, we got the non-buried phase change fibers based on body temperature via melt-spinning, and we estimated that this fiber has a good commercial value for industrial mass production. All the related works are finished and have been applied to the Taiwan and U.S. patent in Dec. 2019. The manuscript is preparing now, and this is a whole story we do for this interesting and valuable works.
- Section 2.3.3 The authors described some special conditions to measure the intrinsic viscosity, but how these conditions were determined was not mentioned; additional details on how those conditions were established should be included in the description of the method.
Response: Thanks for the reviewer’s comments. Section 2.3.3 in the intrinsic viscosity part is modified in “The synthesized PBABI copolyesters were measured in 1.0 g dL-1 within a mixture of phenol and tetrachloroethane in a weight ratio of 60/40 was obtained using an Ubbelodhe viscometer at 25 ±0.05 ℃. The IV of each sample was calculated using the Solomon-Ciuta equation:
Where C is the concentration of the solution; t is the flow time of the sample solution, and t0 is the flow time of the pure solvent. Measurements of each sample were made five times to calculate the averaged IV values and tabulated in Table 2. Please see the revised manuscript.
- Sections 2.3.4 Details on what type and length of column was used were not included; the authors are requested to provide a full description of the method they used for GPC measurements.
Response: Thanks for the reviewer’s suggestions. All the detailed information in GPC was modified in the revised manuscript.
- Page 6 line 195-196 Phrase “The intensity of these two absorption peaks can be enhanced as rasing the of IA concentration, explaining that the IA molecule was successfully copolymerized into a main-chain of PBABI copolyesters” was not clear and it should be reformulated.
Response: Thanks for the reviewer’s comments. The sentence was clarified in “The absorption peaks in 1639 and 817 cm-1 can be enhanced as rasing the of IA concentration, explaining that the IA molecule was successfully copolymerized into a main-chain of PBABI copolyesters.” Please see the revised manuscript.
- The authors did not conclude which one of the four compositions could be used successfully and practically as a smart coating material on 3-D air mesh fabric. This mention was done just within the abstract.
Response: Thanks for the reviewer’s recommendation. As the reviewer said, we only referred it in the last sentence of the abstract; hence, we decided to delete it. Because it is a very long story, as we described in question 6. We focused the results in “the thermal, mechanical properties and crystallization rate of PBABI copolyesters can be tuned through the adjustment of BTCA and IA concentrations.”

Round 2
Reviewer 1 Report
After corrections the manuscript reads well. I suggest publication in Polymers.